# Accumulation of continuously time-varying sensory evidence constrains neural and behavioral responses in human collision threat detection

**Gustav Markkula**[1]*, **Zeynep Uludağ**[2¤], **Richard McGilchrist Wilkie**[2], **Jac Billington**[2]

**1** Institute for Transport Studies, University of Leeds, Leeds, United Kingdom, **2** School of Psychology, University of Leeds, Leeds, United Kingdom

¤ Current address: Department of Experimental Psychology, Ardahan University, Ardahan, Turkey
* g.markkula@leeds.ac.uk

**Data Availability Statement:** The primary research data for this study, as well as the software code implementing the experimental paradigm, data

## Abstract

Evidence accumulation models provide a dominant account of human decision-making, and have been particularly successful at explaining behavioral and neural data in laboratory paradigms using abstract, stationary stimuli. It has been proposed, but with limited in-depth investigation so far, that similar decision-making mechanisms are involved in tasks of a more embodied nature, such as movement and locomotion, by directly accumulating externally measurable sensory quantities of which the precise, typically continuously time-varying, magnitudes are important for successful behavior. Here, we leverage collision threat detection as a task which is ecologically relevant in this sense, but which can also be rigorously observed and modelled in a laboratory setting. Conventionally, it is assumed that humans are limited in this task by a perceptual threshold on the optical expansion rate–the visual looming–of the obstacle. Using concurrent recordings of EEG and behavioral responses, we disprove this conventional assumption, and instead provide strong evidence that humans detect collision threats by accumulating the continuously time-varying visual looming signal. Generalizing existing accumulator model assumptions from stationary to time-varying sensory evidence, we show that our model accounts for previously unexplained empirical observations and full distributions of detection response. We replicate a pre-response centroparietal positivity (CPP) in scalp potentials, which has previously been found to correlate with accumulated decision evidence. In contrast with these existing findings, we show that our model is capable of predicting the onset of the CPP signature rather than its buildup, suggesting that neural evidence accumulation is implemented differently, possibly in distinct brain regions, in collision detection compared to previously studied paradigms.

analyses, and computational models, are available at: https://doi.org/10.17605/OSF.IO/KU3H4.

**Funding:** GM was supported by the Wellcome Trust / University of Leeds Institutional Strategic 549 Support Fund (https://wellcome.org/), grant 204825/Z/16/Z, and the UK Engineering and Physical Sciences Research Council (https://epsrc.ukri.org/), grant EP/S005056/1. JB was supported by the Leverhulme Trust (https://www.leverhulme.ac.uk/), grant RF-2019-343\10. The funders had no role in study design, data collection and analysis, decision to publish, or preparation of the manuscript.

**Competing interests:** The authors have declared that no competing interests exist.

## Author summary

Evidence accumulation models of decision-making propose that humans accumulate noisy sensory evidence over time up to a decision threshold. We demonstrate that this type of model can describe human behavior well not only in abstract, semi-static laboratory tasks, but also in a task that is relevant to human movement in the real world. Specifically, we show that a model directly accumulating the continuously time-varying visual looming (optical expansion) of an approaching obstacle explains full probability distributions of when humans can detect this collision threat. Using electroencephalography, we find indications that this type of evidence is accumulated differently in the brain compared to evidence accumulation in previously studied, more abstract tasks. Our experimental paradigm, model, and findings open for wider application of this type of decision-making model to laboratory and real-world tasks with ecologically relevant, time-varying sensory evidence, and further studies into how such decisions are implemented neurally. There are also societal implications: In applied safety research and traffic accident litigation it is conventionally assumed that human collision detection is limited by a fixed perceptual threshold, an assumption that our results show to be highly inaccurate.

## Introduction

Human decision-making is a long-standing research topic, spanning disciplines such as psychology, neuroscience, economics, and human factors [1–5]. In recent decades, evidence accumulation models (also known as drift diffusion or sequential sampling models) have emerged as one dominant account, positing that decisions are made once noisy evidence has been integrated over time up to a decision threshold [6–11]. These models have been successful at explaining distributions of behavioral choices and response times across numerous laboratory paradigms, e.g., where participants make categorization decisions about ambiguous stimuli, or choose between options with different subjective or objective value [6–11]. There is also strong neurophysiological support for the idea that the brain indeed implements something akin to evidence accumulation in these types of tasks [5, 10–12]. Notably, there is mounting support for the idea that signatures of neural evidence accumulation can be observed using human electroencephalography (EEG), in the form of a centroparietal positivity (CPP) that builds up during deliberation and peaks when the overt response is made [13–19]. However, computational modelling of evidence accumulation decision-making has so far focused on laboratory paradigms using stimuli that (i) have stationary or only intermittently and/or noisily changing saliency over time [7, 18, 20–24], and (ii) are abstract in nature, typically not mapping directly to any real-world task.

It is currently an open question whether decision-making is well-described by evidence accumulation models in less cognitive and more embodied task contexts, relating to human sensorimotor control, movement, and locomotion in the real world. The nervous system performs myriad choices of motor actions to perform tasks such as keeping the body upright [25], balancing a stick [26], intercepting a ball [27] or avoiding collisions with other cars while driving [28]. Do evidence accumulation mechanisms play a role also in these contexts? [29] One challenge in answering this question lies in the nature of the sensory evidence being used: A hallmark feature of real-world sensorimotor behaviors is that they depend on continuously time-varying sensory stimuli, such as joint angles, sight point rotations or optical expansion rates, of which the exact, externally measurable values are important for successfully shaping the behaviour [25–29]. This is in stark contrast with most existing evidence accumulation

modeling work, which has emphasized stationary evidence (in part because this enables computationally efficient model-fitting [30]), with the rate of evidence in the model typically fitted as a free parameter per experimental condition, without a mechanistic link to the properties of the external stimulus. We and others have begun exploring accumulation models of which the input evidence instead scales directly with external sensory data, in tasks such as stick-balancing [31], visual and vestibular judgment of self-motion [32–34], longitudinal and lateral control in car driving [29, 35–38], and road-crossing decisions [39, 40], but these studies have so far not performed model testing and selection at the same level of detail as is typical in the broader evidence accumulation model literature.

Here, we aim to close this gap by developing and studying a paradigm where participants detect onset of visually looming (optically expanding) collision threats. We chose this task because it is an ecologically relevant task with time-varying sensory evidence, and accumulation of visual looming has also been suggested–but not conclusively proven–in several of the mentioned previous studies [36–38], yet this task nevertheless permits collection of large numbers of repetitions in a controlled laboratory environment, enabling detailed model fits of full per-participant probability distributions of response. There are also some specific predictions to test: Conventionally, it is assumed that humans can detect collision threats once the rate of optical expansion of the obstacle's projection onto the observer's retina exceeds a looming detection threshold (LDT) [41–43]. This LDT assumption has been adopted in basic perceptual psychology research into collision avoidance and target interception [44, 45], time-to-contact estimation research [46–48], sports science [49, 50] and applied research in the road traffic safety domain [51–56]. Notably, the LDT assumption is also used in traffic accident litigation, to answer questions about whether an appropriately attentive driver should have been able to avoid a crash [57, 58]. However, some of the early literature on the LDT reported that the kinematics of the collision course (i.e., the movement trajectories of the observer and collision object) could seemingly affect the value of the threshold itself [41, 42, 52]. We have previously proposed that such kinematics-dependencies in human collision detection ability could be understood if, instead of an LDT, collision threat detection were determined by evidence accumulation of the visual looming signal [35], a hypothesis which we test here. We also complement our behavioral observations with concurrent EEG recordings, to investigate whether the previously reported CPP signature could be observed in our paradigm, and if so whether the nature of this neural signature aligned with the predictions of our time-varying evidence accumulation model.

## Results

We conducted an experiment where we simultaneously recorded participants' overt looming detection responses and concurrent EEG. Rather than opting for the type of abstract stimuli conventionally used in neuroscientific research on collision perception [59, 60], to emphasize the connection to real-world collision threat detection we instead chose to create a laboratory version of the driving test track experiment by Lamble et al. [52]. In their experiment, participants followed a lead vehicle at either 20 m or 40 m distance, and pressed their car's brake pedal as soon as they saw the lead vehicle come closer, which it did by a $0.7 \text{ m/s}^2$ deceleration (not accompanied by a brake light signal). In our laboratory implementation, illustrated in Fig 1A and described in detail in Materials and methods, participants were instructed to fixate a location on a screen, where an image of the back of a car appeared at an appropriate optical size for either 20 m or 40 m viewing distance, with minor horizontal and vertical perturbations over time. The car image either remained the same size for 7 s before disappearing (catch trials; 16.7% of the total number of trials) or began, after a random delay in the 1.5–3.5 s range, to

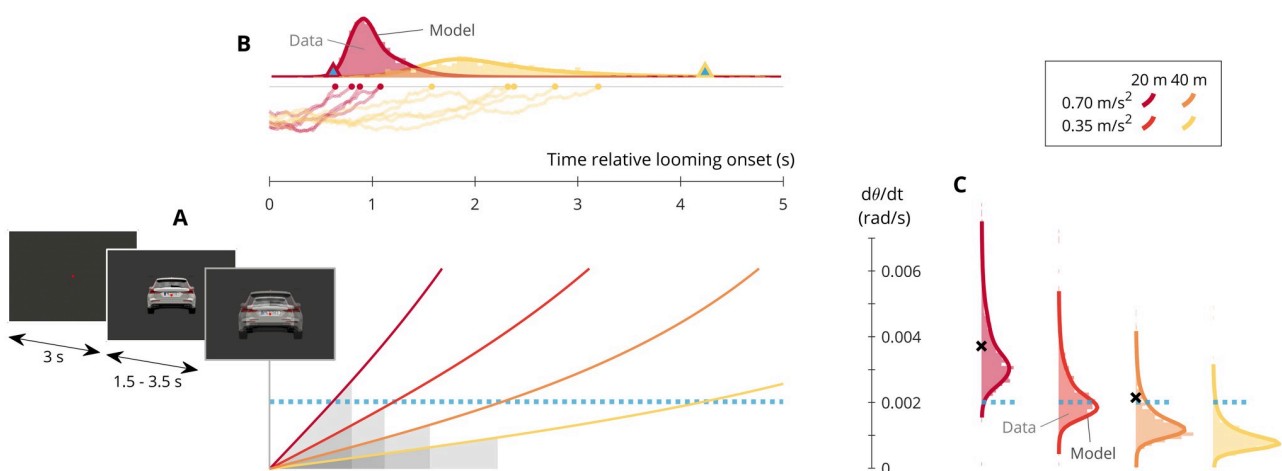

**Fig 1. Overview of paradigm, model, and behavioral results.** (A) In each trial, participants fixated a target, at which an image of the back of a car appeared, and after a variable time delay began to optically expand following one of four different looming trajectories (solid lines; colors per corresponding kinematical conditions as indicated in the boxed legend). The dotted line shows the across-experiment average optical expansion rate $\dot\theta_{mean}$ at which participants reported detection of this visual looming. The shaded regions all have the same area, to illustrate why evidence accumulation predicts detection at lower $\dot\theta$ in conditions where this quantity increases more slowly. (B) Histograms show the participants' detection response times across the entire experiment for the fastest and slowest of the four looming conditions, overlaid with the corresponding predictions from per-participant maximum likelihood fits of the variable-gain accumulator model (model AV; thick solid lines). This model posits that the visual looming evidence shown in panel A is integrated over time together with normally distributed noise, up to a fixed threshold at which detection occurs (example time histories of noisy integration of the looming input shown as thin solid lines, with circle symbols at the fixed decision threshold). The triangle symbols indicate the response times that would be predicted by a conventional looming detection threshold model, with $\dot\theta_{mean}$ as threshold. (C) Optical expansion rates at which the participants (histograms) and fitted accumulator model (lines) reported detection, in the four different looming conditions. The dotted lines again show $\dot\theta_{mean}$, and the black crosses indicate the detection thresholds reported in [52] for the same kinematical conditions.

optically expand, recreating the looming trajectory of a decelerating lead vehicle (i.e., accelerating toward the observer). Participants were instructed to press a key as soon as they saw the car "coming closer"(i.e., growing on the screen). Lamble et al. included also non-foveal detection conditions, which we omitted. Instead, we extended the design by including a 0.35 m/s$^2$ lead vehicle deceleration, for a total of four kinematical trajectories, with distinct profiles of visual looming, as shown in Fig 1A. We denote the projected optical angle of the lead vehicle stimulus on the participant's retina $\theta$, and its optical expansion rate $\dot\theta = d\theta/dt$, increasing nonlinearly with time both because of the vehicle acceleration and because the visual angle of an object is (approximately) proportional to the inverse of its distance from the observer. Note also that in each looming condition there was a direct relationship between response time (the horizontal axis in Fig 1A and 1B and optical expansion rate at response (the vertical axis in Fig 1A and 1C. To align with the existing literature on looming detection, our basic inferential testing on the behavioral data focused on $\dot\theta$ at response, whereas to align with the literature on evidence accumulation modeling, our model-fitting was instead focused on distributions of response time.

## Overt responses refute the fixed looming detection threshold assumption

After exclusion of a small minority of trials for early (0.6%) and missing (0.2%) detection responses, and a larger number of trials for electrooculographic indications of eye blinks (15.9%; see Materials and methods for details), the final data set included 22 participants, with an average of 182 trials per participant (an average of 46 trials per looming condition).

Fig 1C shows that our data replicated the kinematics-dependency reported by Lamble et al. [52], with detection occurring at lower average optical expansion rates $\dot{\theta}$ for the larger initial distance ($F(1, 3698) = 1255.48; p < .0001$). Also in an absolute sense, our $\dot{\theta}$ values at detection were similar to those observed in the test track experiment (black crosses in Fig 1C), but slightly lower, potentially due to the reduced noise in the laboratory environment and the use of a finger key press instead of a foot pedal to report response.

As initially suggested in [35], detection at lower $\dot{\theta}$ values for larger initial distances is predicted by a looming accumulation account, because looming grows more slowly from larger distances, and because accumulation (i.e., integration) of a small $\dot{\theta}$ over a long time is equivalent to accumulation of a large $\dot{\theta}$ over a short time; see the shaded areas in Fig 1A. Similarly, for lower deceleration magnitudes, where looming develops even more slowly, the looming accumulation account also predicts detection at further decreased $\dot{\theta}$ values. This was the motivation for the inclusion of the 0.35 m/s$^2$ deceleration condition, and the observed $\dot{\theta}$ values at detection were indeed further reduced for this lower magnitude of deceleration (Fig 1C; $F(1, 3698) = 810.26; p < .0001$).

These behavioral findings strongly reinforce the idea that looming detection occurs at magnitudes of optical expansion rate that are dependent on the kinematics of the collision course, in contrast with the conventional LDT assumption of a situation-independent threshold for detection. The triangle symbols in Fig 1B indicate the response times that would be predicted by a situation-independent looming threshold fixed at the average $\dot{\theta}$ at detection observed across this experiment. These LDT predictions are too early in fast looming conditions, and too late in slow looming conditions, which is precisely the qualitative pattern of errors that one would expect to see if participants' responses were instead determined by evidence accumulation of optical expansion rate.

From a methodological point of view it is worth noting that our behavioral analyses also identified statistically significant effects of experimental block (slightly increased looming sensitivity in later blocks) and the 1.5–3.5 s pre-looming wait time (slightly increased looming sensitivity with increased pre-looming wait time). These effects were substantially smaller than the effects of looming condition and between-participant differences (see Table B in S1 Appendix), and were therefore not separated out in the subsequent model fitting described below.

## A visual looming accumulator model accounts for full detection distributions

As illustrated in Fig 1B, the looming accumulation hypothesis can be computationally formalized as a single-boundary accumulator (or drift diffusion) model, with its rate of evidence accumulation (sometimes referred to as "drift rate") at each point in time determined by the momentary optical expansion rate, multiplied by some gain, and where overt detection response occurs once an evidence threshold is reached. Noise in the evidence accumulation process (e.g., due to noisy sensory input, interference from other brain activity, or both) gives rise to variability, i.e., probability distributions of response time. It may be noted that our model, like previous evidence accumulation models of detection of intermittent, subtle changes in abstract stimuli [14, 16], effectively implements Page's cumulative sum ('CUSUM') technique for change detection [61].

The conventional LDT assumption is completely deterministic, and as such does not make predictions about probability distributions. However, one might consider a *stochastic threshold model*, positing that looming detection occurs once a noisy optical expansion rate signal first exceeds a fixed threshold. In fact, such a model would also predict the qualitative findings

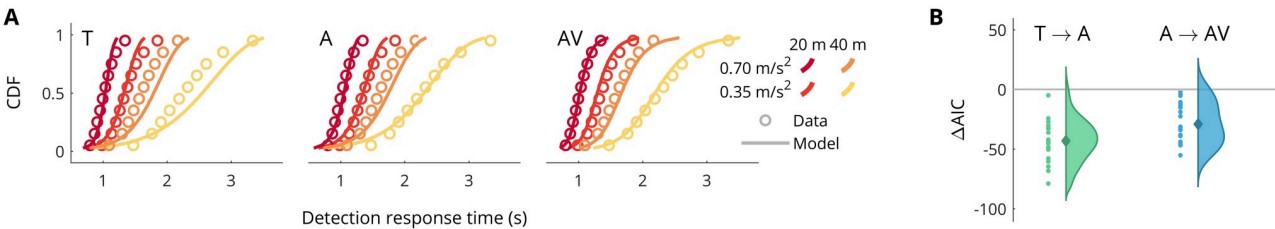

**Fig 2. Model comparisons.** (A) Averaged ("Vincentized") per-participant cumulative density functions (CDFs) of looming detection response time, for human participants and the maximum-likelihood-fitted threshold model (T), accumulator model (A), and variable-gain accumulator model (AV), in the four looming conditions. (B) Per-participant differences in Akaike Information Criterion (AIC) for the T → A and A → AV model comparisons. Negative ΔAIC values indicate preference for the latter model in the comparison.

reported above, since in conditions where looming develops slowly, there would be more time for large noise values to occur by chance, thus eventually exceeding the threshold even if the sensory signal itself is still sub-threshold. We therefore tested also this type of model, and compared it to the accumulator model.

Due to the time-varying drift rate in our accumulator model, there is no closed-form expression for its response time distribution [62]; we estimated these distributions numerically instead. All considered models were fitted per participant, both using maximum likelihood estimation (MLE) and approximate Bayesian computation (ABC), to the per-participant response time distributions in each looming condition. The emphasis in this paper is on the MLE results, whereas the ABC results, mostly reported in the Supporting information, provide additional confirmation of the key conclusions, a more complete view of model parameter estimates, and allowed us to follow up on auxiliary questions which would have been computationally prohibitive under the MLE approach.

Fig 2A shows detection response time distributions, visualized as cross-participant averages (using the "Vincentizing"method [63]) of per-participant data and MLE model fits. The threshold model (model T) does indeed capture the qualitative effect of looming condition, but is unable to accurately reproduce the location and shape of the response time distributions. Similarly to the patterns shown in Fig 1A for the deterministic threshold model, model T has a tendency to predict responses that are substantially too late in slow looming conditions. The accumulator model (model A) does not have this problem, and achieves a noticeably better fit despite having the same number of free parameters as model T.

When modelling perceptual decision-making in paradigms with stationary stimulus saliency, it has often been found that assuming between-trial variability of the stationary rate of evidence accumulation is needed to closely reproduce human response time distributions [10]. Analogously, we investigated an extended version of our basic accumulator model, where the input gain applied to the optical expansion rate was not constant per participant, but instead drawn at random per trial from a normal distribution, of which the standard deviation thus becomes an additional free model parameter. This extended model (model AV) produced distributions that more closely matched those of the human detection responses. Fig 2B shows the relative goodness of fit of the three models, in terms of differences in Akaike Information Criterion (AIC). These results indicate a very strong preference for the accumulator model over the threshold model for all participants but one, with an average ΔAIC of -43.2 (a difference of more than 14 suggests "very strong support" for the preferred model [64]). This analysis also indicated that for most participants, the additional model complexity introduced by the input gain variability (model AV) was warranted given the improvements in model fit (average ΔAIC = -29.1). Per-participant fits for models T, A, and AV are shown in Fig A in

S1 Appendix. Model AV had four free parameters: the non-decision time $T_{\mathrm{ND}}$, the accumulator noise intensity $\sigma$, and mean and standard deviation $K$ and $\sigma_K$ of the looming input gain; estimated values for these parameters across participants are shown in Fig B in S1 Appendix.

The ABC analysis also favored the accumulator model over the threshold model. The geometric mean of the per-participant Bayes factor (an estimate of the expected Bayes factor for hypothetical additional participants [65, 66]) in favor of the accumulator model was 3.0–7.3 ("substantial evidence" [67]), depending on the choice of the ABC distance threshold hyperparameter $\epsilon_{\mathrm{RT}}$, with the highest Bayes factors for the more stringent $\epsilon_{\mathrm{RT}}$ (see Fig E in S1 Appendix). The ABC comparison of accumulator models with and without variable gain was inconclusive (geometrical mean Bayes factor 1.4–1.5 in favor of model A), possibly because of the relatively broad priors we used, which may have excessively penalized model complexity [68].

Using both MLE and ABC methods we also investigated other model variants, incorporating gating of the looming input (requiring it to exceed a minimum threshold before contributing to the accumulation) or evidence leakage (as a form of short-term memory decay), as well as more complex models incorporating different combinations of these various model assumptions. As further described in S1 Appendix, none of these alternative models were found preferable over the variable-gain accumulator model (model AV).

As an additional test of this best-performing model, we also examined its predictions in response to variations in pre-looming wait time. As mentioned above, the model-fitting was blind to this experimental manipulation. However, as shown in Fig C in S1 Appendix, model AV nonetheless predicted the observed pattern of increased looming sensitivity with increased pre-looming wait times, with approximately correct magnitudes.

## Looming accumulation explains onsets of pre-response scalp potentials

Fig 3 illustrates the main EEG findings. The response-locked scalp maps in Fig 3A show a positivity at the overt response, in line with the CPP observed by many others [13–19]. Fig 3B and 3C show stimulus-locked and response-locked ERPs, per condition, averaged over five electrodes centered on Pz. The original paper on the CPP centered its analysis on the CPz electrode location [13], but many subsequent reports have shown more parietally located CPPs, consistent with what we observe here [14–16, 18, 69]. Again in line with previous observations, Fig 3C shows that this positive wave builds up before the overt response and peaks at the response itself, including a characteristic separation at this peak, with higher CPP amplitudes for the more salient looming conditions [13, 14, 16, 18]; see the dots along the bottom of Fig 3C.

However, the CPP we observe here differs in at least two respects from previous observations. First, the build-up of the CPP has previously been reported to be of similar duration as the overt response times, typically only 100–200 ms shorter [13–18]. This is consistent with the idea that the CPP reflects evidence accumulation that starts shortly after stimulus presentation. In contrast, average response times per looming condition in our experiment ranged from 1.1 to 2.4 s, yet it is clear from Fig 3C that the average CPP build-up duration was shorter than 0.5 s for all conditions. Second, since in previous studies CPP increase over ERP baseline has been obvious soon after stimulus onset, conditions with slower responses (typically due to less salient stimuli) have produced CPP profiles with build-up commencing earlier in time before the overt response [13–15, 17]. In contrast, in our response-locked ERP data, there is no obvious separation between conditions in when the CPP build-up commences.

Fig 3D shows that the CPP signal is subtly affected by EEG pre-processing choices (most notably, the 0.1 Hz low-pass filter we used causes a slight suppression to negative voltage

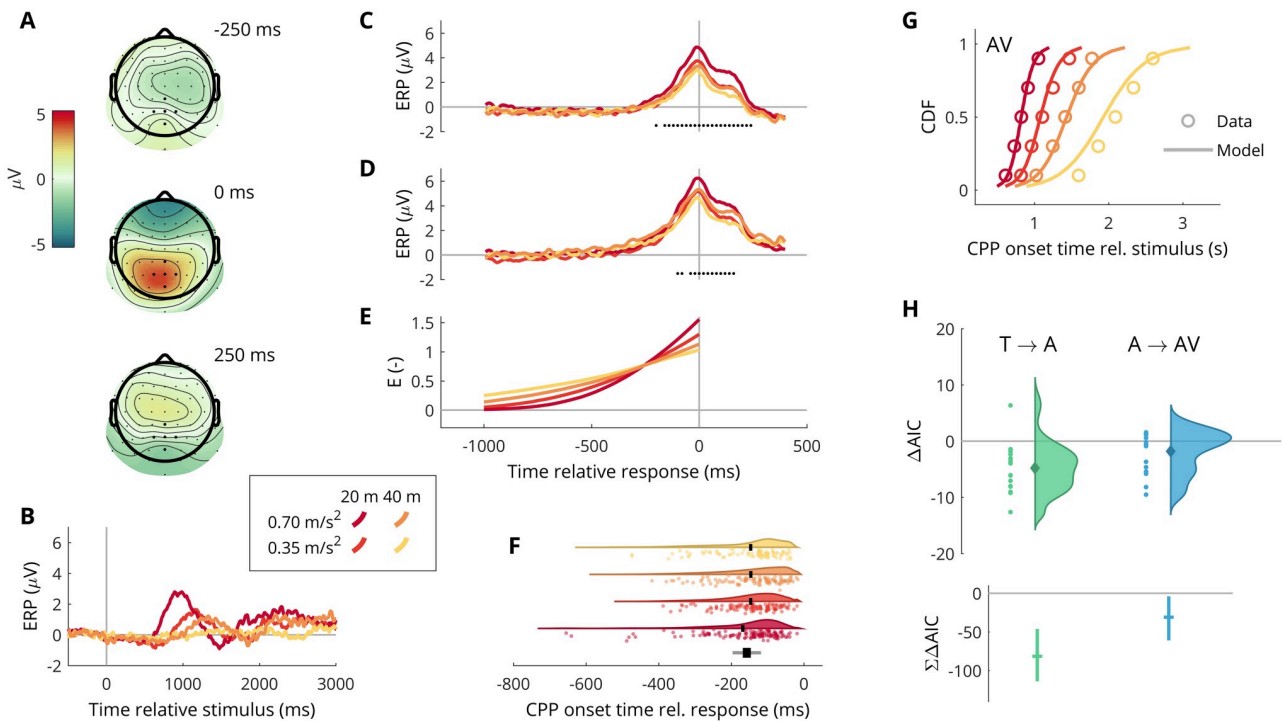

**Fig 3. EEG results.** (A) Grand average scalp potentials shortly before, at, and after overt detection response. (B) Event-related potentials (ERPs) relative to the looming stimulus onset, averaged over five electrodes centered at Pz (marked in A), in the four different looming conditions. (C) The same four ERPs as in (B), but instead response-locked, i.e., relative to the time of overt detection response. The dots along the bottom show where an ANOVA indicated a statistically significant main effect of looming condition on these ERPs. (D) As panel C, but without high-pass filtering and ocular artefact removal. (E) Response-locked average accumulated evidence $E$ in each looming condition, for the best-fitting variable-gain accumulator model AV. Note that the model traces converge at the decision threshold $E = 1$; the exact location of this point in the plot depends on how much of the non-decision time in the model is assumed to be due to sensory and motor delays, respectively. (F) Estimated onsets of the pre-response centroparietal positivity (CPP) relative to the overt detection response, in the four looming conditions. This includes the (17 out of 22) participants for which the CPP onsets could be reliably estimated. The black markers indicate condition means. The intervals plotted in black and gray at the bottom of the panel show, respectively, the maximum observed difference between condition means, and the upper edge of a 95% confidence interval for this difference. (G) Averaged ("Vincentized") per-participant cumulative density functions (CDFs) of CPP onset time relative to the looming stimulus onset, for the participants and the maximum-likelihood-fitted variable-gain accumulator model. (H) The top panel shows per-participant AIC differences for the T → A and A → AV model comparisons, when fitted to the CPP onset data. The bottom panel shows the total cross-participant sums of AIC differences, with 95% confidence intervals.

before the CPP build-up; see further Fig F in S1 Appendix), but the general aspect of a late, condition-independent CPP onset remains. Meanwhile, as can be seen in Fig 3E, the onset of evidence build-up in our behavioral model was both early and condition-dependent, as expected since the model directly accumulates the looming input.

Previous studies have indicated that purely behavioral fits of evidence accumulation models can yield model evidence profiles that align qualitatively with the corresponding CPP [14, 15, 17–19], but as described above this was not the case here. One possible reason for this could be that our models were flexible enough to achieve good behavioral fits for a range of parameterizations, with a range of widely different evidence build-up profiles, and that our purely behavioral fits were therefore not enough to observe an alignment between model evidence and recorded CPP signatures. Therefore, using ABC, we investigated whether fitting the models simultaneously to the behavioral and neural data could identify a behaviorally well-fitting model which did not exhibit early separation of accumulated model evidence, thus aligning better with our CPP observations, but no such model was identified; see Fig J in S1 Appendix.

A different possible explanation for our late, rapid, and seemingly condition-independent CPP could be that (i) the looming accumulation process indicated by the behavioral modeling results is not directly reflected in the CPP, and (ii) the observed CPP instead reflects a second stage of the response decision process, which begins only once the looming accumulation process reaches threshold. To further investigate this hypothesis, we estimated full distributions of CPP onset times from the EEG data, by averaging ERPs over trials with similar response times, to increase signal-to-noise ratio, and identifying the last time where the averaged ERP exceeded 30% of its value at the overt response. We found that this allowed us to reliably estimate CPP onsets for 17 out of the 22 participants. We then investigated our hypothesis using this dataset in two ways: Firstly, as illustrated in Fig 3F, we analyzed the distributions of CPP onset time relative to overt response. In line with our hypothesis that the process reflected in our CPP has a duration that is independent of looming condition, we found that the largest difference in the average CPP onset between two conditions was 22 ms (interval marked in black at bottom of Fig 3F) and that the upper edge of a 95% confidence interval for this difference was 79 ms (interval marked in gray). In other words, any effect of looming condition on CPP onset relative to detection response was small in magnitude (and not statistically significant in our dataset; $F(3, 377) = 1.94; p = .133$). Secondly, we refitted, using MLE, the threshold and accumulator models to the estimated CPP onsets instead of to the overt responses. We found that the accumulator models could account well for the distributions of CPP onset, and reliably better than the threshold model; see Fig 3G and 3H (and see Fig I in S1 Appendix for more detailed results for models T and A). Because the CPP onset dataset being fitted to was reduced compared to the behavioral dataset, a reduced statistical power of the model comparisons should be expected, and the obtained per-participant ΔAIC values (Fig 3H, top) were indeed smaller than for the corresponding behavioral model comparisons. However, for the overall test of whether to prefer A over T across our entire experiment, the total difference in AIC was still very large (-81.4 and -31.0 for the T → A and A → AV comparisons; see Fig 3H, bottom, also showing 95% confidence intervals for these differences). The results illustrated in Fig 3F–3H were robust to variations in EEG pre-processing and our CPP onset estimation method; see further Materials and methods, and Figs F, H, and I in S1 Appendix.

## Discussion

The results presented here support four main conclusions. First, that human collision threat detection occurs at optical expansion rates that are highly dependent on the kinematics of the collision scenario. We replicate the previously observed effect of initial obstacle distance [41, 42, 52], and additionally demonstrate an effect of obstacle acceleration, predicted by our looming accumulation hypothesis. Taken together with the mentioned previous literature, our findings strongly refute the LDT assumption, i.e., the assumption of a single, kinematics-independent threshold for looming detection. As illustrated in Fig 1A, this conventional assumption can yield estimated detection times that are incorrect by several seconds. We therefore caution against further use of the LDT assumption, not least in the applied context of road traffic safety, where it has influenced research, recommendations, and legal proceedings [51–58].

Second, we show that not only the qualitative patterns of kinematics-dependency, but also full probability distributions of collision threat detection can be explained by an evidence accumulation model, assuming that the optical expansion rate information is integrated over time, with noise, up to a threshold at which detection is reported. We thus provide a computational account of how humans perform collision threat detection, which explains both average tendency and precise patterns of variability in performance. From an applied perspective, the

accumulator models proposed here can be considered as an alternative to the LDT assumption. It should be noted, however, that the focus here was on investigating human ability of collision threat detection in a controlled laboratory experiment, rather than to provide and validate a model for applied use. The alignment with the test track findings by Lamble et al. [52] is encouraging, but further real-world validation, ideally covering a more diversified set of kinematical scenarios, would be advisable. For example, while an evidence leakage assumption was not required to account for the data in our experiment, such an assumption might become important in even slower looming conditions, where a model without any memory decay could be overly prone to purely noise-driven detection responses. (See the Supporting information for further discussion of the various alternative model variants tested here.) Another interesting question for future work is whether these improved models of collision threat detection (which as mentioned above effectively implement a change detection algorithm which is optimal under certain assumptions [61]) can support improved models of collision avoidance response. In the road traffic context, some existing models of collision avoidance response suggest that detection and response are separate and sequential steps [57, 58, 70], whereas other accounts suggest that defensive responses are instead driven directly by kinematical urgency, without a clear role for a first, separate step of detection [58, 71, 72].

Third, we provide strong support for the idea that established evidence accumulation models of decision-making can be extended beyond typical laboratory paradigms with static or intermittently changing abstract stimuli, to tasks with ecologically relevant, continuously time-varying sensory evidence, directly using the externally measurable stimulus as an input to the evidence accumulation. We and others have reported that evidence accumulation models show promise for modelling decisions in real-world tasks, e.g., when to apply brakes in response to a developing collision threat [36–38], or on whether and when to cross a road with oncoming traffic [39, 40]. However, in these contexts it has not been possible to fit full response time probability distributions per participant, a minimum expectation in evidence accumulation modeling of more typical, abstract laboratory tasks. Drugowitsch et al. [32, 33] provided compelling support for evidence accumulation decision-making in their visual-vestibular heading discrimination paradigm, but did not emphasize detailed fits of response distributions. Ratcliff and Strayer achieved this level of model-fitting stringency in a driving setting, but did so by using a paradigm of speeded response to discrete stimuli, thus abstracting away from the continuously time-varying nature of the real-world driving task [73, 74]. Our looming detection paradigm was instead chosen to enable similarly rigorous model analyses of an ecologically relevant, continuously time-varying stimulus. It is notable that, among our alternative models, the accumulator model with between-trial variability in input gain (model AV) performed best for a majority of participants. For static input evidence, (i.e., $\dot{\theta} = $ constant, in our case), this input gain variability reduces to between-trial variability in a static accumulation rate, a very common assumption in past modeling work, with much empirical support [10]. Our results thus demonstrate how this model assumption for static evidence paradigms can be usefully generalized to paradigms with time-varying evidence. Our paradigm and models may provide useful starting points for further research into decision making with continuously time-varying evidence, both in other sensory detection tasks (cf., e.g., [75, 76]) as well as in sensorimotor control tasks of basic or applied nature [29, 31, 36–40].

Fourth, in contrast with previous studies on the CPP signature, we show that in our paradigm the late *onset* of the CPP, rather than a build-up rate present from early on after stimulus presentation, can be explained by evidence accumulation. The existing literature features several paradigms that are similar to ours in tasking participants with detecting low-saliency changes in sustained stimuli [13, 14, 16, 18], for example in the form of gradual changes in

visual contrast [13], following time courses not dissimilar to the looming trajectories studied here (Fig 1A). These studies and others have provided converging evidence for the notion that the CPP source (i.e., the neural circuits giving rise to the CPP signature; see [77]) is involved in (or connected to) an early, sustained, and saliency-dependent accumulation of evidence for the decision to respond [13–18]. Our behavioral modeling results support this type of evidence accumulation account of looming detection, yet the CPP in our paradigm is late, rapid, and without a clear effect of stimulus saliency on its duration. We did not hypothesize in advance that our CPP results would differ from previous findings in this way. For this reason, and because the CPP onset analyses we performed here were simple and exploratory in nature, we are unable to draw any firm conclusions about the underlying reasons for the nature of our CPP signatures. However, one seemingly plausible hypothesis would be that a key factor is our use of an ecologically relevant stimulus, specifically visual looming, known to be processed in phylogenetically old subcortical brain structures [78]. Aligning with findings in non-human species [79–82], functional magnetic resonance imaging in humans has implicated structures such as the superior colliculus and the medial pulvinar nucleus of the thalamus in processing of visual looming [60]. These structures play important roles in attentional orientating [83], and have cortical projections circumventing early visual areas, for example to the middle temporal (MT) visual area, known to be involved in processing of motion cues [84]. In a general sense, this difference in connectivity may play a role in why our ERP results stand out. More specifically, both our behavioral and CPP results can be understood if it is assumed that the pathways for looming processing include neural circuits implementing evidence accumulation detection of collision threats, from which only the decision outcome (threat detected or not) is communicated onward to the CPP source, which then carries out a rapid, second-stage evidence accumulation, implementing the higher-level, modality-general decision of mapping stimulus to response in the task at hand [13]. This tentative 'two-accumulator' hypothesis would explain why the onset distributions of the late and rapid CPP in our data can be well accounted for by a looming accumulation model. In the Supporting information we provide a computational formulation of this hypothesis and illustrate how it might explain also the at-response CPP separation between looming conditions (Fig 3C and Fig K in S1 Appendix). The two-accumulator hypothesis is interesting not least in light of findings that the CPP correlates with subjectively reported experience of the perceptual decision being formed [69]. From this perspective, the late CPP signatures in our data suggest the empirically testable hypothesis that visual looming evidence accumulation (before CPP onset) occurs with near-zero subjective awareness or confidence. This would align well with conventional notions of an early perceptual limitation on looming detectability, but recasting the limitation as an evidence accumulation decision process instead of a perceptual threshold.

## Materials and methods

### Ethics statement and open software/data

All procedures were approved by the School of Psychology Research Ethics Committee, University of Leeds, reference number PSC-484. The primary research data for this study, as well as the software code implementing the experimental paradigm, data analyses, and computational models, are available here: https://doi.org/10.17605/OSF.IO/KU3H4.

### Experimental design

The objective of the experiment was to observe participants' detection of a visually looming object, to what extent this detection was influenced by the kinematical details of the object's

approach, and whether traces of the response process could be observed in participant scalp potentials.

The basic task was a computer-simulated replication and extension of the foveal looming conditions of the test track experiment in [52]. The paradigm was implemented in MathWorks MATLAB using PsychToolbox v3.0.14 [85, 86]. The stimulus was a photographic image of the back of a 1.85 m wide and 1.43 m high passenger car (used with permission from Volvo Car Corporation), as shown in Fig 1A. This image was displayed over a dark gray color on a 24 inch (0.53 m × 0.30 m) 60 Hz TFT screen at 1920 × 1080 pixels resolution. The original image was at higher resolution than shown on screen, and was scaled to appropriate size and displayed with antialiasing using the OpenGL trilinear filtering provided by PsychToolbox.

A central fixation target (a red dot, diameter 6 pixels, 0.095 degrees visual angle) was displayed throughout each experimental block. In each trial, initially only this fixation target was shown for 3 s, then the stimulus image appeared centrally on the screen, accompanied by an auditory tone, displayed at a size corresponding to an initial distance of either 20 or 40 m (subtending 5.30 and 2.65 degrees horizontal visual angle, respectively). Some trials were catch trials without any looming, at which the stimulus remained at the same size for 7 s before disappearing. In non-catch trials, the stimulus remained at the same size during an initial pre-looming wait time, one of 1.5, 2, 2.5, 3, 3.5 s, whereafter the size of the stimulus was gradually increased to reproduce the looming visual input from a car decelerating at either 0.35 or 0.7 m/s$^2$; as shown in Fig 1A. The participants were instructed to keep their eyes on the fixation target, refrain from blinking while the car was being shown and until their response, which they were instructed to give by pressing the space bar on a computer keyboard with their right hand "as soon as you see the car coming closer, in other words when it is growing on the screen". Trials terminated once participants either (a) made a correct looming detection response, after which the stimulus continued looming for another 0.5 s before disappearing, to avoid the impact of this visual transient interfering with the EEG measurements at time of response, (b) made no response before the looming stimulus had reached a trial expiry threshold of 0.03 rad/s (about ten times the threshold typically stated in the literature), or (c) made an incorrect, early detection response before the onset of visual looming; in this last case a distinct auditory tone was played to inform the participant of their incorrect detection response.

The participants viewed the stimulus screen at a distance of 1.00 m, meaning that each screen pixel subtended a visual angle of 0.95 arcmin (0.016 degrees), lower than the 1.6 arcmin threshold reported in [87] for maximum Vernier acuity with antialiased stimuli. To further reduce the risk of pixel effects, and to mimic the conditions of the replicated test track experiment [52], the stimulus was displayed with small horizontal and vertical oscillatory perturbation throughout, generated by moving the simulated viewport as if the participant themselves were sitting in a car, with perturbation spectra based on measurements from real driving; see Table A in S1 Appendix for details.

## Procedure

Participants provided written informed consent before taking part in the experiment, which was carried out in a dark room with the participant sitting in front of the stimulus display, supported by a chin rest. In a first demonstration block of four trials, the experimenter demonstrated the task, including the auditory tone given upon incorrect, early responses, as well as a feedback screen that was shown after each block. This feedback screen listed average response times and frequency of correct responses for all blocks so far, and encouraged participants to rest if their response times were increasing. The participants then decided themselves when to start the next block. The participants first completed a practice block of 12 trials, two of each of

the four looming conditions (2 initial distances × 2 acceleration levels), and four catch trials. Then followed the five experimental blocks, each with a total of 48 trials, eight catch trials and ten repetitions of each of the four looming conditions (two repetitions for each of the five pre-looming wait times), making for a total of 5 × 40 = 200 looming trials per participant, 50 for each looming condition. Trial order was fully randomized per block and participant.

## Participants

The target initial sample size was twenty-five participants, to provide a comfortable margin over the total number of trials collected in previous studies reporting on the CPP [13, 14]. Twenty-six right-handed participants were recruited from a local pool of participants, all with normal or corrected-to-normal vision, and with no history of psychiatric diagnosis, severe brain injury, motor diseases or any skin conditions. EEG recording was incomplete for one participant, and data from the remaining twenty-five participants, of ages between 20 and 46 years (mean 26.5), 12 male and 13 female, were retained for further analysis.

## Data acquisition and preprocessing

Behavioral responses were recorded at the 60 Hz refresh rate of the display screen. Out of the 25 × 40 = 1000 catch trials, there were 61 (6.1%) with false detection responses. The catch trials were not further considered in the analyses or modeling. Out of the 25 × 200 = 5000 trials with looming stimuli, participants responded before looming onset in 32 trials (0.6%), and non-responses were observed in 8 trials (0.2%); these early and non-responses were included in the behavioral model fits, but were excluded from all EEG analyses.

EEG data were recorded at 1024 Hz, using a 64 electrode 10–20 international cap Biosemi system. Electro-oculogram (EOG) electrodes were placed above and below the left eye and at the outer canthus of each eye. EEG preprocessing was done using EEGLAB v14.1.1 [88], first resampling to 512 Hz, then using the PREP pipeline EEGLAB plugin v0.55.3 [89] for robust re-referencing to average channel and interpolation of noisy channels. PREP interpolated one of the five channels analyzed as part of the pre-response positivity analyses here (Pz and surrounding channels CPz, POz, P1, P2) for only three participants (one of which was later excluded due to ocular artefacts; see below), in each case only one of the five channels was interpolated. Then, bandpass filtering was done using EEGLAB's sinc FIR filter with a Kaiser window, with pass/stopband ripple of 0.001, lowpass filter of with 45 Hz cut-off and 5 Hz transition bandwidth, and high-pass filtering at 0.1 Hz cut-off with 1 Hz bandwidth. Following [13], trials with pronounced ocular artefacts were rejected by the vertical EOG difference exceeding 100 $\mu$V, excluding 796 trials (15.9%). 395 of these trials were from three specific participants, who therefore failed to reach a minimum of at least 30 trials in each looming condition. These three participants were excluded from further analysis. Further ocular artefacts were identified and removed from the EEG data per participant, using EEGLAB's independent component analysis (ICA) functionality. The final dataset included 22 participants with a total of 4013 looming trials, i.e., an average of 46 trials per participant and looming condition. For event-related potential (ERP) analysis, the EEG data of looming trials were divided into epochs from 1 s before to 8 s after the looming onset in each trial (sufficient to include all responses in all conditions given the 0.03 rad/s trial expiry threshold mentioned above), and the EEG data for each epoch were baseline-corrected, using the average of the last 200 ms before the looming onset as baseline. Then, the five channels centered on Pz mentioned above were averaged to yield the final signal used in the CPP analyses (per-condition averages across participants shown in Fig 3B and 3C; per-condition averages per participants shown in Fig G of S1 Appendix). The analysis illustrated in Fig 3C was also rerun after disabling the 0.1 Hz EEG high-pass

filter and the ICA ocular artefact removal, confirming that neither of these two preprocessing steps substantially altered the obtained CPP signatures; see Fig 3D and Fig F in S1 Appendix.

For the analyses which focused specifically on the onset of the CPP, we increased the signal to noise ratio by sorting trials on response time per participant and looming condition, separating the sorted trials into groups of five trials, and taking the average response-locked ERP within each such group. We then identified the CPP onset for each averaged trial as the last sample where the averaged response-locked ERP was less than 30% of its value at the overt response. We excluded averaged trials where this did not occur within 1 s before the response, or where the ERP at response was less than +2 $\mu$V, resulting in a total exclusion of 284 (37.4%) out of the 760 averaged trials. 134 of these exclusions were due to five participants, who distinguished themselves from the rest either by having no clear ERP peak at response (Cohen's $d < 0.3$ when comparing ERP between 0.5 s before response and at response) or by the at-response peak occurring at close to zero voltage; see Fig G in S1 Appendix. These five participants were excluded from the CPP onset analyses, leaving a data set of 17 participants and 445 averaged trials (an average of 26 per participant). Since our CPP onset estimation method was novel, and not originally planned for, we also conducted sensitivity analyses. As illustrated in Figs F and H of S1 Appendix, these analyses showed that the obtained CPP onset estimates were robust to variations in the parameters of our method, but also that our method was not suitable for use on non high-pass filtered ERP data, which would have been desirable since the high-pass filtering we used subtly altered the CPP signal (cf. Fig 3C and 3D). Future work should improve on these onset estimation methods, for example along the lines of the approach in [90].

## Statistical analysis

To test for effects of the kinematical looming conditions on the optical expansion rate at behavioral detection response, we carried out a repeated-measures ANOVA, implemented using MATLAB's anovan function with participant as a random factor, independent variables initial car distance (2 levels) × acceleration magnitude (2 levels) × experimental block (5 levels) × pre-looming wait time (5 levels), limited to first-order interactions only, and $\log(\dot{\theta})$ as the dependent variable; see Table B in S1 Appendix for full results. To study separation between response-locked ERPs for different looming conditions (Fig 3C and 3D), we performed ANOVAs at 20 ms intervals, with the response-locked ERP as the dependent variable, participant as a random factor, and looming condition (4 levels) as independent variable. We also performed this type of ANOVA to test for an effect of looming condition on CPP onset relative response (Fig 3F). The 95% confidence intervals in Fig 3F and 3H were obtained by 100,000 sample bootstraps from the empirical data in question.

## Computational models

The optical expansion rate accumulator models investigated here can be described by the following discrete update equation for the accumulated evidence $E$ at time step $i$:

$$E(i) = \max(0, E(i-1) + \tilde{K}\dot{\theta}(i)\Delta t + \sigma v(i)\sqrt{\Delta t}), \qquad (1)$$

where $\tilde{K}$ is an accumulation gain parameter either drawn at random per trial from a normal distribution $\mathcal{N}(K, \sigma_K^2)$ with $\sigma_K$ as a free parameter (model AV), or kept constant per participant ($\sigma_K = 0$; model A), $\Delta t$ is the discrete simulation time step length, and the final term is a discrete implementation of a Wiener process with noise intensity $\sigma$, with $v(i)$ drawn at random per time step from a standard normal distribution $\mathcal{N}(0, 1)$. Note that we included a reflecting

lower boundary at zero (the max function), as often done for evidence accumulators with a single decision boundary [7, 20, 91, 92]. The optical expansion rate $\dot{\theta}(i)$ is the time derivative of the optical size of the collision obstacle (here, the lead vehicle):

$$\theta(i) = 2\arctan\frac{W}{2D(i)} \Rightarrow \dot{\theta}(i) = \frac{WV(i)}{D^2(i) + W^2/4} \qquad (2)$$

where $W$ and $D(i)$ are the width of and momentary distance to the collision obstacle, and $V(i)$ is the momentary speed at which it is coming closer [28]. The accumulator model makes a detection decision once $E(i) \geq 1$ ($E$ is in arbitrary units, so one of decision threshold, $K$, or $\sigma$ can be fixed without loss of generality) overtly responding a non-decision time $T_{ND}$ later. A fraction $\alpha_{ND} = 0.3$ of $T_{ND}$ was assumed to occur before the evidence accumulation, based on [29, p. 189]; this value did not affect the behavioral model fits or comparisons, only model evidence visualizations like the one in Fig 3E. For a description of the other accumulator model variants that were also tested, see the Supporting information.

The stochastic threshold model investigated here (model T) can be written on a similar form:

$$E(i) = \frac{\dot{\theta}(i)}{\dot{\theta}_d} + \sigma v(i), \qquad (3)$$

again with overt detection response a time $T_{ND}$ after $E(i) \geq 1$, i.e., the parameter $\dot{\theta}_d$ is the looming detection threshold parameter. Note that the estimated value of $\sigma$ for this model will depend on the discrete simulation time step length; we used $\Delta t = 0.02$ s across all models. The models were simulated from the start of each trial, i.e., the pre-looming wait time was also simulated, and the accumulator models were initialised at $E = 0$ for each trial.

## Model fitting by approximate Bayesian computation

Our main goal of ABC was parameter estimation rather than model selection, so we used relatively broad, non-informative priors. To make the obtained Bayes Factors reasonably meaningful, and for increased computational efficiency, we identified approximate limits for the model priors by means of a first ABC fit to the information available from the previous test track experiment [52]; see Supplementary material for details. We then fitted each model variant (T, A, AV, etc) to each participant separately. To implement ABC rejection sampling [93, 94], for each parameterization sampled at random from the prior distribution, we generated a simulated data set of the exact same nature and size as the data obtained from one human participant, (same number of repetitions, pre-looming wait times, etc.) and calculated and stored a set of twenty summary statistics; the response time quantiles {0.1, 0.3, 0.5, 0.7, 0.9} for each of the four looming conditions separately. We then compared the simulated RT quantiles to each human participant's data, for each participant retaining only parameter samples where all twenty absolute differences between simulated and observed RT quantiles were below a rejection threshold $\epsilon_{RT}$. These retained samples provide an approximation of the posterior parameter distribution for the participant [93, 94]. There are several more advanced versions of ABC than this basic rejection sampling algorithm, but this method was preferred here because it allowed us to obtain individual per-participant fits without extra simulations of the model (which is the computationally costly step), and because it made computationally feasible the investigations illustrated in Fig E of S1 Appendix, showing that the ABC model comparisons were robust to the choice of the hyperparameter $\epsilon_{RT}$. In the Supporting information we also describe how we used ABC to jointly fit both our behavioral and neural data, finding that

excessive model flexibility was not the reason for the mismatch between our CPP observations and our behavioral models' evidence traces.

## Model fitting by maximum likelihood estimation

Maximum likelihood estimation of model parameters was carried out using exhaustive grid search over the model parameter ranges identified by the abovementioned initial ABC fits to the previous test track experiment, with each parameter's range uniformly divided into 20 searched grid values. For each parameterization in this grid, as in the ABC fits a replica of the real experiment was simulated, but upscaled by a factor 20 to 1000 trials per looming condition, yielding a numerical response time distribution per condition, estimated at a bin size of 0.25 s. For these fits, all models were also extended by assuming a probability $P_C = 0.01$ (our conclusions were robust to variations in this value) per trial of 'contaminant' responses poorly described by our model, e.g., due to temporary lapses in participant attention [95], modeled as a uniform distribution across the time range from looming onset to trial expiry. (Whereas our ABC fitting method is robust to such contaminant responses, they can disrupt the MLE fits if they fall in low probability response time bins, which may occasionally be numerically estimated to zero probability by the contaminant-free model.) Likelihoods were then estimated from the resulting numerical probability distributions, per participant and model parameterization. This model fitting method was computationally feasible for models with up to four free parameters (T, A, AV, AG, AL); based on the results from the ABC fits we would not expect to see substantial further improvements with the more complex models.

## Supporting information

**S1 Appendix. Additional method details, analyses, and results.**
(PDF)

## Author Contributions

**Conceptualization:** Gustav Markkula, Richard McGilchrist Wilkie, Jac Billington.

**Data curation:** Gustav Markkula.

**Formal analysis:** Gustav Markkula.

**Investigation:** Zeynep Uludağ.

**Methodology:** Gustav Markkula, Zeynep Uludağ, Richard McGilchrist Wilkie, Jac Billington.

**Writing – original draft:** Gustav Markkula.

**Writing – review & editing:** Gustav Markkula, Zeynep Uludağ, Richard McGilchrist Wilkie, Jac Billington.

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
