## [Decision Letter · Decision Letter 0]

19 Dec 2020

Dear Prof. Markkula,

Thank you very much for submitting your manuscript "Accumulation of continuously time-varying sensory evidence constrains neural and behavioral responses in human collision threat detection" for consideration at PLOS Computational Biology.

As with all papers reviewed by the journal, your manuscript was reviewed by members of the editorial board and by several independent reviewers. In light of the reviews (below this email), we would like to invite the resubmission of a significantly-revised version that takes into account the reviewers' comments.

All reviewers indicate they find this an interesting paper, with an interesting application of accumulator models to a real-life situation. They also all have some concerns. I agree with these concerns, which in my main boil down to the following issues that need to be addressed:

1) to what extent is this really a change detection problem rather than a two-alternative forced choice decision, and how does this impact the modeling? I would like to see, if possible, the suggested single-threshold model Reviewer 1 is suggesting

2) exactly how do the situational variables map onto the model parameters, and could other mappings also be possible, as Reviewer 2 is suggesting?

3) is it possible to estimate the CPP onset with certainty, as Reviewer 1 asks? And could the CPP become more "normal" when a different high-pass filter is used, as Reviewer 3 suggests?

4) a final, more theoretical problem, related to issue 1), is how does the accumulator model "know" when to start?

I invite you to submit a revised version of the manuscript in which these issues, as well as the issues raised by the reviewers are addressed. And for now, happy holidays!

We cannot make any decision about publication until we have seen the revised manuscript and your response to the reviewers' comments. Your revised manuscript is also likely to be sent to reviewers for further evaluation.

Sincerely,

Marieke Karlijn van Vugt, PhD

Associate Editor

PLOS Computational Biology

Wolfgang Einhäuser

Deputy Editor

PLOS Computational Biology

All reviewers indicate they find this an interesting paper, with an interesting application of accumulator models to a real-life situation. They also all have some concerns. I agree with these concerns, which in my main boil down to the following issues that need to be addressed:

1) to what extent is this really a change detection problem rather than a two-alternative forced choice decision, and how does this impact the modeling? I would like to see, if possible, the suggested single-threshold model Reviewer 1 is suggesting

2) exactly how do the situational variables map onto the model parameters, and could other mappings also be possible, as Reviewer 2 is suggesting?

3) is it possible to estimate the CPP onset with certainty, as Reviewer 1 asks? And could the CPP become more "normal" when a different high-pass filter is used, as Reviewer 3 suggests?

4) a final, more theoretical problem, related to issue 1), is how does the accumulator model "know" when to start?

I invite you to submit a revised version of the manuscript in which these issues, as well as the issues raised by the reviewers are addressed. And for now, happy holidays!

Reviewer's Responses to Questions

**Comments to the Authors:**

Reviewer #1: The authors examine a model of collision threat detection by humans. They contrast an evidence accumulation model of this capability against a simple threshold model without evidence accumulation. The evidence accumulation style of model wins hands down. That model comparison seems quite compelling. They also examine the centroparietal positivity (CPP) computed from EEG recordings of the participants during task performance. In their work, the CPP does not appear to reflect an evidence accumulation process, but rather something closer to decision readout perhaps. This is in contrast to the results of Kelly & O'Connell and colleagues, who established a relationship between evidence accumulation and the CPP. The stimuli here (looming vehicles), however, might not be processed in quite the same way as less naturalistic stimuli. Still, there was a CPP, and the absence of correlation with evidence accumulation is noteworthy.

I found the paper to be interesting, novel, and of obvious practical importance. I have a few larger and more minor points of concern.

Major issues:

1) The experimental situation isn’t just different from standard perceptual tasks like dot motion discrimination in terms of the nature of the stimulus, but also in terms of the types of choices. This task does not involve two choices. It involves only one, and the question is when to choose it. This is more like a change-detection task than the two-choice type of task usually addressed by the diffusion model, isn't it? How does a change-detection model compare to the diffusion model? I believe change-detection algorithms such as the CUMSUM test are accumulators too, but without a lower threshold. �This is not necessarily a bad thing, but it just makes the mapping to the decision literature a little less straightforward. Isn't this really a change-detection problem? I think the authors ought to consider that.

2) Further, there are applications of single-boundary diffusion models to, among other things, interval timing (Simen et al., 2016). The nice thing about a single boundary model is that it has a very simple, closed-form expression for the first-passage time (or decision time) distribution, which is the inverse Gaussian distribution. This is much easier to compute than the first-passage time distribution of a two-boundary model. Much faster than approximate Bayesian computation too. It might be worth fitting to the data.

3) It's not clear to me how the authors embedded the assumption of a declining threshold within trials into their model fits. What was the shape and equation for that function? On p. 13, the text suggests there was not in fact a declining threshold. It was fixed at 1. So how does that square with the description in Fig. 1? The text should describe how the declining threshold was modeled, or why it wasn't, given Fig. 1.

4) Need a table of the model parameters and their fitted values and any other info needed (hyperparameters?). I think one should always display that in publications that use this type of model.

5) I'm concerned about excessive reliance on the *onset* of CPP activity as the main dependent variable in the EEG section of the paper. Maybe that’s not the best feature of the CPP to examine, particularly when it would seem that onset-estimates might be sensitive to the way in which they are computed (?). It seems from the figure that buildup rate and maximum amplitude clearly distinguish the conditions. It would help to know more details about the behavior (average RT) to compare to the CPP data. �

Minor points:

1) Fig. 1, B: the time axis isn't labeled under the response time histograms. Please label

2) p. 4: Define \\dot{\\theta}. It is used before it's defined. Please define first

3) p. 5: The statement about "bias" is confusing, I think. Drift is indeed related to evidence accumulation in typical applications of the diffusion model, But "bias" is usually considered a different form of shifting the evidence accumulation process up or down at the start of a trial. It could occasionally be said that the drift is biased, but it sounds odd to me to treat bias and drift as the same thing. I think users of the diffusion model would agree.

4) p. 6: Wow, I did not expect to see evidence in favor of auditory over combined auditory and visual

5) p. 7, Plese define "onset" here. How was it computed?

6) I find the word “stringent” to be somehwat unusual. I notice that it sounds unfamiliar

to me from the perceptual decision literature. Maybe “powerful” would actually be a better word? The model is powerful because the tests of its predictions are powerful in the statistical sense of power to discriminate between hypotheses.

7) p. 9: "we show that in our paradigm the onset of the CPP, rather than its build-up profile, can be 303 explained by evidence accumulation". It didn't seem to me that the authors investigated the build-up profile of the CPP that closely. Fig. 3C also seems to show something like differences in the CPP buildup profile, to my eye.

8) p. 10: I like the ending of the Discussion. It seems that this paper calls for a controlled comparison between say, dot motion discrimination, and looming automobile detection. The authors ought to consider doing that comparison in the future. It would really inform the discussion around the CPP and what it reflects.

Reviewer #2: This paper describes a nice proof of concept for an early stage evidence accumulation process that depends on optical expansion and later stage decision process that is measured by CPP. My one major caution for the conclusions in the paper is that they result from highly flexible models (so goodness of fit is not necessarily an indicator of identifying a useful model of the data) and that the evidence of the distinct stages is still dependent on the assumptions that went into the analysis. I include a few minor comments below.

I would appreciate a bit more clarification on the variables in the study. I suggest making explicit the relationship between response time average optical expansion rate. Also, clarify why that was the independent variable rather than response time.

Deceleration speed and initial distance were crossed, but there was no indication that their interaction was tested statistically. Why was that test not included?

Figure 1: Although there is a lot already crammed in, there still needs to be axis labels for each subfigure.

To what extent is the estimation procedure for the CPP onset novel? There was not a clear justification for this approach in the paper. Was this a planned analysis? How does the uncertainty in the estimates from this procedure influence the downstream analyses (i.e., the accumulator model fits).

It may not be appropriate for this journal, but I suggest adding more detail on the cautionary note about applying the LDT to real-world situations. Is the way LDT used conservative enough that the error does not matter? How would the authors suggest improving the approach?

Pg 10, ln 341: It is not clear how the two stage accumulation process implies that visual looming accumulation occurs without awareness. Please clarify.

Pg 11, ln 382: How much data was removed? Do the timeouts vary across levels of the independent variable?

Pg 31, ln 450: Issue with the less than sign.

Reviewer #3: In this paper, Markkula et al use a combination of computational modeling and human electrophysiology to examine the mechanisms of looming decisions, in a naturalistic task of detecting when a car in front has begun to decelerate. The results show quite convincingly that there is a role for accumulation of optical expansion rate over time in these decisions, where previous research – and policy - has assumed that there is a fixed threshold set on momentary optical expansion rate. They also find that during this task, a centro-parietal ERP signal (CPP) associated with evidence accumulation is strongly present, but, interestingly, has a much more short-lived temporal extent and seems to begin rising a fixed time before the response is made regardless of the strength of the evidence on which the decision is based. The conclusion is thus that evidence accumulation is involved in the looming decision but that in this context, the CPP reflects a second stage of processing after accumulation has reached threshold. I find this an excellent paper, clearly written and with thorough and innovative methods, on a very interesting question that has clear real-world relevance. My comments mostly seek clarifications on how the models were structured and whether the most important alternative accumulation models have been ruled out, none of which harm the central conclusion that these decisions do involve some form of accumulation over time and not simply a fixed looming threshold.

My first question may simply be out of ignorance of the looming field. But it struck me that what most decision makers might do in this situation – because it seems the more natural thing we do when driving a car – is set a threshold on proximity (a translation of size into an estimate of number of metres away), rather than a rate of optical expansion, because if the car is far enough in front, a very steep rate of optical expansion might not tend to call for braking just yet. I wondered more generally why the “evidence” in this scenario isn’t proximity rather than optical expansion rate, and if the decision is re-cast as the former, does that obviate the accumulation since distance is the integral of speed? If my comment is off the mark, it might nevertheless serve to highlight what more general journal audience might think when looking at the situation.

The most puzzling aspect of the results for me was that the best quantitative fit to the behavioural data did not seem to require either leakage or a threshold set on the evidence. Either of these mechanisms would be able to explain a short-lived evidence accumulation process as suggested by the CPP, and it also seems that either of them is required to fully account for how decisions can be made under such gradual and temporally-uncertain conditions. That is, the winning model has to assume that evidence accumulation begins some fixed time after the onset of the target regardless of the strength of that target, which almost assumes the brain precisely knows the timing of onset in a way that it couldn’t possibly, given the timing jitter. A threshold on the evidence would provide a very simple mechanism for knowing when to kick off the accumulation process, rather like Purcell and Schall’s gated accumulator model, and leakage would obviate the need for the accumulator to be kicked off at all – it would just be continuous. For both of these mechanisms, the decision process could be modelled from the beginning of the stimulus and allow naturally for the targets appearing at variable times, and potentially also any effects of that timing (which were not discussed in the paper – were later targets detected differently than earlier ones? See Boubenec et al 2017). But was either model implemented in this way? I also wondered about the impact of constraining evidence to be positive (“E(i) ≥ 0”), which crops up in the supplemental information (top p4) but not in the main paper and should be more fully explained (ideally, in the mathematical equation, e.g. by using a half-wave rectification operator if that is what is going on). If evidence samples are not permitted to be negative, then the accumulation of noise alone would build up and eventually cross threshold, because positive values are added to the total but negative ones ignored. Might this contribute to the simulated timecourse of accumulation being longer than it would otherwise be? And there is no provision for the inevitable noisiness of the crossing of a sensory threshold, for instance in a nondecision time variability parameter or by implementing the threshold not on the pure physical signal but the noisy evidence representation itself, with noise sigma and all. My overall point is that given the ability of threshold and leak models to provide a fuller explanation of performance of the task starting from stimulus outset, as well as a short-lived CPP, the details and justification for the specific ways these models were implemented are very important to lay out fully, with perhaps a discussion of alternative ways (e.g. NOT rectifying E(i)).

Also related to the remarkably short-lived CPP, I wondered what possible impact the fairly strong high pass filter (0.1 Hz cutoff) may have had. This would have the effect of reducing slow shifts in the signals, of the kind that the model predicts but are missing in the real CPPs. It is therefore important to explore the impact of removing the high pass filter.

More minor:

Drugowitsch et al (2014; eLife) examined decisions about time-varying evidence and might be worth a citation given the authors’ framing in terms of breaking away from stationary evidence

line 83 is written as if it is assumed the reader is already familiar with Lamble et al. I also suggest finishing the intro or starting the results with a brief intro to the physics of the situation for the uninitiated, e.g. define theta!

I suggest briefly stating in the main part of the results, perhaps in the figure 1 legend, that misses and false alarms were so few as to be not worth showing, because otherwise readers may wonder if the behavioural data are incompletely shown, until they reach the methods, where it is stated in an odd place, in the EEG preprocessing section. This is the same place we learn what proportion of trials were catch trials, which is a detail I believe should be in the results where the task is described as it can be integral to a subject’s responding strategy. I may have missed it, but I did not see a statement of the proportion of catch trials that resulted in false alarms, and whether those were included in modelling?

In Fig 3C: I suggest adding a note to the legend to explain that the splaying-out of the model simulation’s traces across conditions before the response time is a result of post-decision accumulation and that the time of coalescence some ~300 ms before the response marks the point of threshold crossing in the model. Otherwise I think it may be confusing for some readers that the model has a constant threshold yet the simulated traces reach different levels at response time, until they have sifted through the methods.

Line 227: the “last time ... exceeded?” Is this a typo – should it be the last time 30% was NOT exceeded?

The term “pre-decision” is used several times to refer to, e.g. , the response-locked CPP, but I think this will be quite confusing for most, especially when most of the CPP seems to rise after the timepoint the model indicates as the threshold crossing. Perhaps stick to “response-locked,” “pre-response” or “pre-commitment”

**Have all data underlying the figures and results presented in the manuscript been provided?**

Reviewer #1: Yes

Reviewer #2: Yes

Reviewer #3: Yes

PLOS authors have the option to publish the peer review history of their article (what does this mean?). If published, this will include your full peer review and any attached files.

Reviewer #1: **Yes: **Patrick A. Simen

Reviewer #2: **Yes: **Joseph W. Houpt

Reviewer #3: **Yes: **Simon Kelly
---

## [Decision Letter · Decision Letter 1]

26 Mar 2021

Dear Prof. Markkula,

Thank you very much for submitting your manuscript "Accumulation of continuously time-varying sensory evidence constrains neural and behavioral responses in human collision threat detection" for consideration at PLOS Computational Biology. As with all papers reviewed by the journal, your manuscript was reviewed by members of the editorial board and by several independent reviewers. The reviewers appreciated the attention to an important topic. Based on the reviews, we are likely to accept this manuscript for publication, providing that you modify the manuscript according to the review recommendations.

Thank you very much for your thoughtful revisions. Two reviewers have now accepted your manuscript. Reviewer 3 still has one concern about the filtering of the ERPs, and I concur. I encourage you to submit a revision that addresses this concern. Many thanks in advance!

Sincerely,

Marieke Karlijn van Vugt, PhD

Associate Editor

PLOS Computational Biology

Wolfgang Einhäuser

Deputy Editor

PLOS Computational Biology

[LINK]

Thank you very much for your thoughtful revisions. Two reviewers have now accepted your manuscript. Reviewer 3 still has one concern about the filtering of the ERPs, and I concur. I encourage you to submit a revision that addresses this concern. Many thanks in advance!

Reviewer's Responses to Questions

**Comments to the Authors:**

Reviewer #1: I am pretty satisfied with the changes the authors made. First, I want to acknowledge that they were correct in their responses about three mistakes I made in my original review. And then I have one more concern prompted by another reviewer's comments, and one more suggestion about highlighting the optimality properties of CUSUM, and perhaps their own model.

First mistake: I understand now that the evidence/drift is non-constant over time, so there are no closed-form response time distributions.

Second mistake: I'm not sure how I managed to mistake A and AV as standing for Audio and Audio-Visual -- I must've been reviewing another paper with audio and visual conditions at the same time and confused them! Maybe using "VA" instead of "AV" would've prevented that misreading on my part. That's a trivial change that could reduce the chance of other readers making the same silly mistake. It also has the same order as the corresponding words in "variable-gain accumulator".

Third mistake: Regarding my misinterpretation about whether there were declining thresholds in the evidence accumulator model -- I had to reconstruct why I thought that. It was because of the caption to Fig. 1, parts B and C, in which different final \\dot(\\theta) values at the time of responding are noted -- these final values decrease for slower looming stimuli. Of course, the time-integral of \\dot(\\theta) need not be lower when there are lower final \\dot(\\theta) values. So there's no problem -- I was just confused again. However, since I did make that mistake of interpretation, maybe other readers would too, and it might be worth emphasizing in the caption that the top panel of 1B shows that the integral of the \\dot(theta) values in the bottom panel do indeed reach the same fixed threshold at decision time.

New concern, but not a serious one: I think it's interesting, on my second consideration of this manuscript and of the other reviews, that conceptually, integrating the time-derivative of radial image size over time, without leak, from the start of the trial up to now, is equivalent at every moment to computing the difference between the current radial size and the initial radial size, by the fundamental theorem of calculus. That's only important in the following sense: I can either physically compute a radial velocity, and also compute a running integral of that velocity, or I can take the current size and subtract off my memory of the initial size at the start of the trial. Right? It might seem simpler to some readers to just directly subtract two quantities rather than differentiate, and then undo the differentiation by integrating; and why would the brain use calculus operations when it could just use addition and subtraction?

Personally, I suspect we DO need to compute time derivatives and integrals for all sorts of other reasons, so it's arguably parsimonious for the brain to use differentiation and integration operations. More importantly, the fatal flaw of the pure addition/subtraction approach is that you need to remember an initial radial size, potentially for quite a long time. The velocity computation, in contrast, can be done with a much shorter-duration memory requirement, as in an Euler-method approximation of the derivative -- for that, I just need to remember the radial size from a little while ago in order to subtract it from the current size. And the integration part is really just addition anyway. This might be worth mentioning, if it isn't already mentioned (I couldn't find it).

Regarding the reference to Page's CUSUM procedure for change point detection: it would be interesting to analyze more formally to what extent this model implements CUSUM, as has been done for two-boundary drift-diffusion and the sequential probability ratio test. That's not necessary for this paper. However, as I read the description of CUSUM, something, often the likelihood of the data sample (I think?), is always subtracted off the next data sample before it is added to the running sum. This sounds like a leak term, but I guess it could also just be approximately implemented with a constant inhibition of the accumulator.

Finally, when mentioning CUSUM, I think it would be well worth highlighting the optimality property of CUSUM for certain kinds of scenarios, in terms of minimizing the detection delay of a maximally costly change, while also minimizing the cost of false alarms. Sounds perfect for a case in which the maximally costly change is a deadly collision from failure to detect a vehicle's rapid approach. So there's a reasonable chance that the model here is essentially optimal for maximizing the chance of survival, without also leading to slamming on the brakes every five feet.

As far as the CPP onsets are concerned, I think the new wording in the paper handles that issue, but I would defer to Reviewer 3 on that.

Reviewer #2: The revised article reads well and is generally quite clear. The authors have done well addressing my concerns regarding the earlier submission. I apologize for the independent/dependent mix up and now I am going to have to admit making the mistake to my intro psych class.

Reviewer #3: The authors have provided clear and thoughtful replies to my comments and have made excellent amendments to the manuscript.

On the model specification, it is now clearer how and why it was implemented, and with these clarifications I am satisfied that there aren't obvious alternatives that should have been visited.

On the high-pass filter, however, while I do agree that the waveforms are similar to look at when it is removed, I do think it noteworthy that the first half of the timeframe is less flat, and may contain a subtle buildup aspect. A 0.1 Hz low cut off tends to be about as high as you can go without distorting regular, rapidly-fluctuating ERPs, so it stands to reason that it might begin to subtly distort signals as slow as those involved in the current task. I suspect that if you were to apply a similar 0.1 Hz filter on the simulated waveforms from the model, it may appreciably alter the waveforms in a way that makes the buildup more local to the response - not enough to bring it into alignment with the CPP, perhaps, but in principle, it doesn't seem that the comparison can be fully fair when the CPP is subjected to a transformation that attenuates gradual aspects when the simulated waveforms are not.

I therefore suggest replacing the CPP waveforms and onset estimates in the main figure with the unfiltered ones, or at the very least, computing and showing the unfiltered CPP onset estimates similar to main figure 3D, in supplementary Fig S6A.

Minor:

The abstract says "the model explains CPP onset rather than buildup" but it will not be clear what that means without getting further into the paper - consider whether a tweak here can help to convey that result more clearly, e.g. "The model's estimated decision times align with CPP onset?"

**Have all data underlying the figures and results presented in the manuscript been provided?**

Reviewer #1: Yes

Reviewer #2: Yes

Reviewer #3: None

PLOS authors have the option to publish the peer review history of their article (what does this mean?). If published, this will include your full peer review and any attached files.

Reviewer #1: **Yes: **Patrick Simen

Reviewer #2: **Yes: **Joseph Houpt

Reviewer #3: **Yes: **Simon Kelly

Figure Files:

Data Requirements:

Reproducibility:

References:

---

## [Editor Report · Decision Letter 2]

19 May 2021

Dear Prof. Markkula,

We are pleased to inform you that your manuscript 'Accumulation of continuously time-varying sensory evidence constrains neural and behavioral responses in human collision threat detection' has been provisionally accepted for publication in PLOS Computational Biology.

Best regards,

Marieke Karlijn van Vugt, PhD

Associate Editor

PLOS Computational Biology

Wolfgang Einhäuser

Deputy Editor

PLOS Computational Biology

Congratulations! Your paper has been accepted. Based on my reading, you have satisfactorily addressed the comments of all reviewers.

---

## [Editor Report · Acceptance letter]

22 Jun 2021

PCOMPBIOL-D-20-02020R2 

Accumulation of continuously time-varying sensory evidence constrains neural and behavioral responses in human collision threat detection

Dear Dr Markkula,

I am pleased to inform you that your manuscript has been formally accepted for publication in PLOS Computational Biology. Your manuscript is now with our production department and you will be notified of the publication date in due course.

With kind regards,

Olena Szabo
